# IL-20 Activates ERK1/2 and Suppresses Splicing of X-Box Protein-1 in Intestinal Epithelial Cells but Does Not Improve Pathology in Acute or Chronic Models of Colitis

**DOI:** 10.3390/ijms24010174

**Published:** 2022-12-22

**Authors:** Md. Moniruzzaman, Kuan Yau Wong, Ran Wang, Hamish Symon, Alexandra Mueller, M. Arifur Rahman, Sumaira Z. Hasnain

**Affiliations:** 1Faculty of Medicine, The University of Queensland, Brisbane, QLD 4072, Australia; 2Immunopathology Group, Mater Research Institute—The University of Queensland, Translational Research Institute, Brisbane, QLD 4102, Australia; 3Australian Infectious Disease Research Centre, University of Queensland, Brisbane, QLD 4072, Australia

**Keywords:** IL-20, epithelial cells, colitis, goblet cells, ERK1/2, endoplasmic reticulum stress, IL-22

## Abstract

The cytokine Interleukin (IL)-20 belongs to the IL-10 superfamily. IL-20 levels are reported to increase in the intestines of Ulcerative Colitis (UC) patients, however not much is known about its effects on intestinal epithelial cells. Here, we investigated the influence of IL-20 on intestinal epithelial cell lines and primary intestinal organoid cultures. By using chemical-induced (dextran sodium sulphate; DSS) colitis and a spontaneous model of colitis (*Winnie* mice), we assess whether recombinant IL-20 treatment is beneficial in reducing/improving pathology. Following stimulation with IL-20, intestinal primary organoids from wild-type and Winnie mice increased the expression of ERK1/2. However, this was lost when cells were differentiated into secretory goblet cells. Importantly, IL-20 treatment significantly reduced endoplasmic reticulum (ER) stress, as measured by spliced-XBP1 in epithelial cells, and this effect was lost in the goblet cells. IL-20 treatment in vivo in the DSS and *Winnie* models had minimal effects on pathology, but a decrease in macrophage activation was noted. Taken together, these data suggest a possible, but subtle role of IL-20 on epithelial cells in vivo. The therapeutic potential of IL-20 could be harnessed by the development of a targeted therapy or combination therapy to improve the healing of the mucosal barrier.

## 1. Introduction

Cytokine-based immune therapies are becoming a popular option in the treatment of ulcerative colitis (UC). Several antibodies are currently in use or in clinical trials to reduce inflammation, promote epithelial wound healing and barrier integrity, ultimately to improve the severity of UC. Among others, anti-TNF-α therapy has been found to be the most effective [1]. However, all the current therapies have severe side effects and/or the patients do not respond to the therapies. Therefore, there is still an unmet need to develop new, more effective, universal treatment strategies to cure UC.

Interleukin (IL)-20 belongs to the IL-20 subfamily and IL-10 superfamily of cytokines, and acts through the heterodimeric receptor complexes including IL-20RA/IL-20RB and IL-22RA/IL-20RB [2]. The receptor subunits that can be found in different tissues, however, are abundantly expressed in the skin, pancreas, intestine, and reproductive organs [3]. It has been found that the expression of *IL-20RA*/*IL-20RB* receptor subunits was elevated in the colon during active UC, and expression was further enhanced during remission [4]. The levels of IL-20 were also shown to be higher in active UC compared to patients in remission and healthy controls, however, its impact on disease modulation is still unknown [5]. In various tissues, IL-20 activated a number of signalling pathways, where activation of STAT3 was the most prominent signalling found in the stably transfected HEK293 cells with *IL-20RA* and *IL-20RB* and in the HT-29 intestinal epithelial cells [6]. IL-20 is well-known for regulating skin inflammation, cell differentiation and growth [7,8].

Monocytes, keratinocytes, macrophages, and dendritic cells are the main sources of IL-20 [5]. Despite the pro-inflammatory role in psoriasis, IL-20 was found to promote epithelial wound healing of the injured mouse cornea [9] and foot ulcer in the diabetic *db*/*db* mice [10]. We have previously shown that IL-20 did not affect the colonic epithelial cell line proliferation [11]. Although IL-20 possesses a higher affinity with IL-20RA/IL-20RB than IL-22RA/IL-20RB, IL-20-induced wound healing in *db*/*db* mice was mediated through the later complex [10]. This suggests that depending on a given context, the cytokine-receptor affinity changes. Colonic biopsy samples from the active UC patients showed high *IL-20* expression, which decreased during remission [4]. In addition, decreased frequencies of IL-20 gene polymorphisms (*rs2981573* and *rs2232360*) were also reported to have a positive correlation with the development of UC [12]. These indicate the possible potential role of IL-20 in active colitis, which also could be beneficial in exogenous delivery to promote fast recovery.

However, no studies have yet demonstrated the exact role of IL-20 on intestinal epithelial cells in controlling intestinal inflammation. Therefore, this study has been undertaken to understand the signalling cascades activated and the transcriptomic modifications governed by this cytokine in the primary intestinal epithelial cells from wild-type animals and *Winnie* mice that develop spontaneous colitis. We next evaluated whether IL-20 treatment impacts acute and spontaneous intestinal inflammation using different pre-clinical models of ulcerative colitis.

## 2. Results

### 2.1. Recombinant IL-20 Activates ERK1/2 in Intestinal Epithelial Cells but Does Not Modulate Signaling in Goblet Cells

To understand the effects of IL-20 in the intestine in a controlled manner, we used the primary intestinal organoid culture system. The primary intestinal epithelial cells (IECs) were passaged at least 4–5 times to ensure the absence of immune cells. Comprehensive analyses of signalling pathways in undifferentiated IECs from wild-type (WT) animals showed that recombinant IL-20 (rIL-20) treatment only activated the ERK1/2 pathway (Figure 1A). To assess whether IL-20 activates ERK1/2 in colitis, we utilized the *Winnie* mouse model of colitis, which carries a missense *Muc2* gene mutation which results in an epithelial cell defect. rIL-20 also activated ERK1/2 in the *Winnie* IECs (Figure 1B).

Goblet cells are the major secretory cells in the intestine and are responsible for producing mucins that are the main macromolecular component of the secreted mucus layer. Therefore, *WT* and *Winnie* IECs were differentiated into goblet cells using the notch γ-secretase inhibitor, DAPT. Interestingly, IL-20 did not further activate any of the signalling pathways assessed in the differentiated cells from *WT* or *Winnie* animals, suggesting that it mainly affects epithelial cells (Appendix A).

### 2.2. Recombinant IL-20 Treatment Did Not Modulate Pathology in Acute Chemical-Induced Colitis

In this study, 2.5% dextran-sodium sulphate (DSS) was administered to WT animals in drinking water for 7 days, causing significant body weight loss (Figure 2A), diarrhea, and bloody stools (Figure 2B), which are the hallmarks of this chemical-induced colitis model. Animals were treated with 100 ng/g of body weight (intraperitoneally) of rIL-20 daily for 7 days. Mouse rIL-20 treatment did not alter body weight or diarrhea scores (Figure 2A,B). Colon weight/length ratio is a measure of pathology and demonstrated an increase with DSS administration. However, rIL-20 treatment did not markedly alter colon weight length ratio (Figure 2C). Histological assessment of the colon using H&E staining and periodic acid–Schiff–Alcian blue (PAS/AB) staining for goblet cells, demonstrated the increase in immune cell infiltration and the loss of goblet cells with DSS administration (Figure 2D). However, no major changes were apparent in immune cell infiltration and goblet cell loss with rIL-20 treatment.

### 2.3. IL-20 Is Capable of Reducing Endoplasmic Reticulum Stress; However, Has No Major Effects on Pathology in the Acute DSS-Induced Colitis Model

We have previously demonstrated that IL-22, which is from the same sub-family as IL-20, can reduce cellular stress in intestinal epithelial cells and goblet cells [13]. To confirm that IL-20 similarly affects cellular stress, we utilised the LS174T cell line. rIL-20 reduced the baseline levels of sXBP1 in LS174T cells (Appendix A). The co-treatment of undifferentiated LS174T cells with tunicamycin (Tm) for 6 h and rIL-20 (100 ng/mL) showed that rIL-20 significantly suppressed *sXBP1* gene expression (Appendix A). DAPT treatment of cells, which induces goblet cell differentiation, resulted in a slight increase in *sXBP1*, fitting with previous reports [14]. rIL-20 treatment did not affect the expression of *sXBP1* in DAPT-differentiated secretory LS174T cells, mirroring the ERK1/2 activation observed in epithelial cell organoids, which was lost with differentiation (Figure 1; Appendix A).

Next, we assessed whether there were any changes in the immune cells in the mesenteric lymph nodes (mLNs) with rmIL-20 treatment. Flow cytometric analyses on the mLNs, harvested from the rIL-20 treated *WT* mice, demonstrated that there were no major changes in the CD4^+^ T cells, CD44^+^ activated T cells or α4β7^+^ activated T cells (Figure 2E). However, rIL-20 treatment significantly reduced the number of F4/80^+^ macrophages, suggesting a slight reduction in macrophage infiltration to the mesenteric lymph nodes. Overall, no major changes were observed in the intestinal expression levels of cytokines, including *Il1beta*, *Il6*, *Il17a* and the chemokine *Mip2a*, reflecting the lack of changes in the T cell profile (Figure 2F; Appendix A).

Despite the effects of IL-20 in vitro, there were no major changes in cellular stress markers (*Grp78, sXbp1, Nos2*) observed in vivo in the DSS animals treated with rIL-20 (Figure 2G). Atoh1 (Atonal homolog 1) is a transcriptional regulator of intestinal progenitors, towards secretory cell lineages. We observed a significant increase in *Atoh1* in LS174T cells treated with rIL-20, along with a decrease in *Hes1*, suggesting that IL-20 has the potential to promote goblet cell differentiation (Appendix A). Corroborating the in vitro data, an increase in *Atoh1* was observed with rIL-20 treatment in vivo, however no major changes were observed in the secretory cell markers, antimicrobials *Reg3γ/β* or the major intestinal mucin, *Muc2* (Figure 2H). Gene expression data, along with the histology, suggested that rIL-20 treatment only slightly affected the pathology in the intestine.

### 2.4. IL-20 Treatment Transiently Improves Diarrhea in Winnie Mice without Long-Term Improvements in Pathology

*Winnie* mice provide a spontaneous colitis model, in which progressive pathology develops with age. To determine whether rIL-20 can halt the progression of colitis, we utilized *Winnie* mice at 6 weeks of age as colitis emerges in these animals. We did not observe any major changes in the body weight of the animals treated with rIL-20 (Figure 3A). *Winnie* mice have diarrhea, which progressively worsens with age. There was a transient but significant improvement in diarrhea in the *Winnie* animals treated with rIL-20 (Figure 3B). Despite this, we observed no major changes with rIL-20 treatment in the colon weight/length ratio, histological assessment of pathology, immune cell infiltration, goblet cell volume or expression of secretory cell markers (Figure 3C-I). Gene expression analyses showed that no major changes were observed in the ER stress marker Grp78 and cytokine *Il1-beta* (Appendix A). Interestingly, rIL-20 treatment in *Winnie* mice resulted in a slight decrease in the activated macrophages (MHC II^+^ F4/80^+^ cells), which was accompanied by a significant decrease in *Il6* in the distal colon (Appendix A).

## 3. Discussion

This study aimed to understand the potential signalling mechanism(s) modulated by IL-20 in the intestinal epithelial cells. In addition, we evaluated the therapeutic potential of using recombinant IL-20 in the acute and spontaneous models of colitis. It is interesting that, although IL-20 is in the same subfamily as IL-22 and share common receptor subunits IL-22RA1, these cytokines have vastly differing roles. Depending on the tissue environment and site of action, IL-20 has been described to act as a pro-or anti-inflammatory cytokine, however its role in the intestine is not well-defined.

The receptor subunits for IL-20 include IL-22RA1, IL-20RA, and IL-20RB, and are highly expressed in the colon. It has been found that IL-20 activated the JAK1/STAT3 signalling pathway in the HT-29 intestinal epithelial cells but did not modulate cell proliferation [15]. This is in agreement with our previously published work, as IL-20 did not influence cell proliferation in immortalized intestinal cell lines [11]. However, in this study, we utilized primary murine organoids to isolate the direct effects of IL-20 on healthy epithelial cells, in the absence of immune cells. We show that IL-20 significantly activated ERK1/2 in undifferentiated mIECs from wild-type animals. We observed slightly higher activation of ERK1/2 in Winnie mice, suggesting that the receptor expression may be increased. Previous reports have demonstrated that the epithelial cell expression of IL-20RB and IL-20 protein are increased in colitis [4].

A high DSS concentration in drinking water, given during the acute DSS model, damages the intestinal epithelial lining by stripping off the mucins and depleting goblet cells. This allows luminal bacteria and the associated antigens to penetrate the mucosa and initiate inflammatory immune responses in the epithelium and underlying tissues [16]. Using the DSS model, we did not observe any major improvements in pathology with IL-20 treatment. Similarly, no major changes in pathology were observed in the *Winnie* mouse model of spontaneous colitis. Interestingly, IL-20 treatment increased the expression of *atonal homolog 1* (*Atoh1*) in the DSS model. Although the Atoh1 increase with IL-20 exposure was not observed in the Winnie animals, we confirmed that IL-20 can drive Atoh1 directly using intestinal cell lines. The increase in *Atoh1* was accompanied by a decrease in *Hes1*. *Atoh1* is a transcriptional regulator of goblet cell differentiation, whilst *Hes1* drives the intestinal stem cells towards an absorptive lineage, like the enterocytes [17,18]. This data suggests that IL-20 may drive the transit amplifying cells in the intestine towards a more secretory cell phenotype. This effect of IL-20 could be explained by the lack of receptor expression on goblet cells; IL-20 failed to activate signalling pathways in intestinal organoids after differentiation into goblet cells. Moreover, reports of IL-20 receptors increasing in colitis are accompanied by a loss of goblet cells, suggesting that absorptive epithelial cells are the main target of IL-20, potentially along with immune cells.

IL-20, unlike IL-22, has also been reported to act on immune cells. In particular, IL-20 activated p38 MAPK in monocyte-derived dendritic cells enhanced the expression of co-stimulatory molecule CD86, to increase the migration of DCs [19]. IL-20, through IL-20RA, was also shown to inhibit IL-17A production by γδT cells [20]. We did not note many alterations in the immune cell profile or cytokines, with rIL-20 treatment in our models. There is a possibility that the lack of impact on pathology in vivo is due to non-specific binding or low local concentrations of the cytokine with an intraperitoneal route of administration. However, we did note a decrease in the percentage of F4/80 macrophages in the DSS animals treated with rIL-20. In *Winnie* mice with emerging colitis, whilst no changes in the percentage of F4/80 cells was observed, there was a trend towards a decrease in MHCII+ve F4/80 macrophages with rIL-20 treatment, suggesting a reduction in macrophage activation. The decrease in *Il6* expression in the distal colon of *Winnie* mice with IL-20 treatment could be due to a decrease in activated macrophages.

ER stress and the unfolded protein response are major contributors in IBD pathogenesis. Using the *Winnie* mice, we have previously shown that cellular stress and protein misfolding can prime an immune response even in the absence of microbiota [13,21,22,23,24]. Therefore, therapeutics that can alleviate mucosal ER stress could be of major benefit in healing the barrier in IBD. Using LS174T cells, we found that IL-20 can suppress spliced-Xbp1 from baseline levels or tunicamycin-induced spliced-Xbp1. This is reminiscent of the suppressive effects of IL-10 and IL-22 on tunicamycin induced cellular stress [13,21,25]. In contrast to IL-10 and IL-22, which can reduce cellular stress in goblet cells, the effect of IL-20 was lost when organoids were differentiated into goblet cells. Overall, our data suggests that the observed effects of IL-20 were mediated via the epithelial cells, or potentially the immune cells as reported before [19]. Whilst IL-20 may reduce cellular stress in epithelial cells, it will not act to promote the restoration of the mucosal barrier. Therefore, it will be unlikely that rIL-20 treatment alone is sufficient to improve pathology in intestinal inflammation. Further studies with a targeted delivery of higher doses and/or frequency of the cytokine and antibody, in combination with other immunotherapies, might provide beneficial effects in intestinal epithelial inflammation.

## 4. Materials and Methods

### 4.1. Chemicals and Reagents

DMEM-F12, Advanced DMEM, DPBS, FBS, TrypLE^TM^ Express, pen-strep, glutamax, and protease inhibitor were purchased from Gibco Life Technologies (Grand Island, NY, USA). TGF-β and ROCK inhibitors were procured from Tocris Bioscience (Bristol, UK). Mouse rIL-20 was procured from BioNovus Life Sciences (NSW, Australia). All phosphorylated antibodies against STAT1, STAT3, STAT5, Akt, 42/44MAPK (ERK1/2), NF-κBp65, 90RSK, and c-Jun, secondary antibodies against rabbit and mouse, were obtained from Cell Signaling Technology (Danvers, MA, USA) and β-actin from Novus Biologicals (Littleton, CO, USA). Phosphatase inhibitor PhosSTOP was purchased from Roche (Basel, Switzerland). Antibodies against mouse CD3, CD45, and F4/80 were procured from BD Bioscience (CA, USA), where mouse specific CD4, CD11b, CD11c, CD44, α4β7 were found, and Zombie GreenTM fixable viability kit were from Biolegend (CA, USA).

### 4.2. Animals

In this study, 5–6-weeks-old C57BL/6 male mice were purchased from the Animal Resource Centre, Australia. The animals were housed under standard laboratory conditions maintaining 12 h light/dark cycle, 25 ± 2 °C temperature, and 55–60% of relative humidity, with a standard chow diet and water ad libitum. All experimental protocols conducted in this study were approved by the Institutional Ethics Committee of The University of Queensland (MRI-UQ/107/16/ECR/PDRF/NHMRC).

### 4.3. Murine Primary Intestinal Epithelial Cell Culture (mIECs)

The mouse colon was removed, longitudinally opened, and washed with ice-cold PBS containing 1% pen-strep to remove all faecal material. Tissue was then cut to the size of approximately a centimetre and treated with ice-cold 8 mM EDTA/PBS, containing 1% pen-strep for 1 h at 4 °C. Subsequently, the tissue was incubated with 2 mg/mL collagenase and 50 μg/mL gentamycin in F12-DMEM, containing 10% FBS, 1% Glutamax, and 1% pen-strep (washing medium) for 5–10 min at 37 °C. Crypts were then isolated using a 10 mL medium through 30 s vortexing and repeating the process 5 times. Isolated crypts were collected through washing and centrifugation at 500 rpm for 5 min at 4 °C, and plated in a 24-well plate following Matrigel embedding [26].

### 4.4. Western Blotting

The undifferentiated and DAPT-differentiated mIECs were treated with recombinant IL-20 for an hour and harvested with RIPA buffer (50 mM Tris-HCI, pH 7.5, 150 mM NaCI, 1.0 % Nonidet P-40, 0.1% sodium deoxycholate), supplemented with phosphatase and protease inhibitors. Protein concentrations in the lysates were determined using a BCA assay kit (Thermo Scientific) and samples were stored at -80 °C until use. In total, 40–45 μg of proteins from each sample were resolved using NuPAGE 4–12 % Bis-Tris protein gels (Invitrogen^TM^; Seventeen Mile Rocks, QLD, Australia) at 100 V and transferred to a PVDF membrane using a Thermo Scientific^TM^ iBlot^TM^ 2 dry blotting system. The membranes were then blocked in an odyssey buffer for 2 h. Following incubation with primary antibody overnight at 4 °C, membranes were washed with PBST and treated with a fluorophore-conjugated secondary antibody for 2 h at room temperature. After washing the membranes, images were scanned using the Odyssey Imaging System and processed, and densitometric analysis was conducted on blots with positive bands using Image Studio Lite software (LI-COR Biosciences, Lincoln, Nebraska, USA).

### 4.5. Quantitative Reverse Transcriptase Polymerase Chain Reaction (qRT PCR)

The quantitative reverse transcriptase polymerase chain reaction (q-RT PCR) was performed according to the protocol described previously [25]. Briefly, after desired treatment, cells were lysed using TRIzol^TM^ (Invitrogen) and pure RNA were collected using ISOLATE II RNA Mini Kit from Bioline (Alexandria, NSW, Australia). For the animal samples, tissues were homogenized using beads (Lysing Matrix D Bulk; MP Biomedicals, Seven Hills, NSW, Australia) in TRIzol^TM^ and then the kit instructions were followed. Equal 1 μg of RNA were then used to synthesize the corresponding cDNA, using a Bioline cDNA synthesis kit. Depending on the targeted genes, the cDNA was diluted up to 1:10 ratio to perform PCR. 2.5 μL of diluted cDNA, 0.75 μL of desired primer (Appendix A, 3.75 μL of SYBR green (SensiFAST^TM^ SYBR^®^ Lo-ROX kit, Bioline), and 0.5 μL of DNase and RNase free water were mixed together and run in a Real-Time PCR System (Applied Biosystems^®^ ViiA^TM^ 7, Life Technologies Corporation, CA, USA) for 40 cycles. The Ct values were then analysed using a ViiA 7 software (Life Technologies Corporation). The relative quantitation was determined by the ΔΔCt method, normalized to the housekeeping gene *β-actin* and expressed as a fold difference to the mean of the relevant control samples. The amplification of the targeted genes (primers in Appendix A) was verified from the melting curve obtained from the experiment. The fold-changes were analysed and plotted using GraphPad Prism software (version 6).

### 4.6. Effects of rIL-20 in Acute DSS-Induced Colitis

In C57BL/6 mice, acute colitis was induced by administering 2.5% *w/v* of dextran sodium sulphate (DSS; M.W. 40,000 g/mol; PanReac AppliChem, Barcelona, Spain) in drinking water for 7 consecutive days. Recombinant mIL-20 (100 ng/g daily) was given through intraperitoneal injection starting from the same day of DSS, where the naïve and DSS only groups received PBS equivalent to their body weight (1%). Mice were monitored daily and scored for their body weight change, rectal bleeding, and stool consistency (0 = hard, 1 = soft but form in a shape, 2 = soft but form in shape and falls apart when picked up, 3 = no form, 4 = watery). At day 7, mice were euthanised through cervical dislocation. The colon was removed, and the entire length was measured from proximal colon (PC) to rectum. Colons were then longitudinally opened, cleaned, and their weight recorded. The proximal and distal colons were separately collected for RNA extraction and rolled to fix in 10% formalin for histology. Moreover, the mesenteric lymph nodes were collected to check the immune cell profiles through flow cytometry analysis.

### 4.7. Effects of rIL-20 in Winnie Spontaneous Colitis

Six-weeks-old *Winnie* mice were treated with recombinant mIL-20 (100 ng/g every 2 days; i.p.) for two weeks. Then the animals were sacrificed, the colon tissues were collected and harvested for histology, and qRT PCR analyses were conducted, as described previously.

### 4.8. Flow Cytometry Analyses of the Immune Cells

The collected mesenteric lymph nodes (MLNs) were crushed through a 70 μm strainer and suspended in DMEM containing 10% FBS, 1% glutamax, and 1% pen-strep in ice. The single cells were collected through centrifugation at 500 rpm for 5 min at 4 °C and resuspended in MACS buffer (Miltenyi Biotech, Bergisch Gladbach, Germany). Then, the cells were incubated with fluorophore conjugated primary antibodies specific to CD3, CD4, CD11b, CD11c, CD44, CD45, F4/80, MHC-II, and α4β7 in MACS buffer for 30 min. Cells were then washed and specific cell types were identified through Frotessa (BD Biosciences, CA, USA). Data were analysed using FlowJo (Ashland, OR, USA).

### 4.9. Histological Analysis of Intestinal Tissues

Intestinal tissues were folded as Swiss roll and immediately perfused in 10% formalin and incubated at room temperature for 24 h. Tissues were subsequently embedded in paraffin and sectioned as per requirements. The tissue was sectioned at 5 μm and stained with hematoxylin-eosin (H&E) or periodic acid–Schiff–Alcian blue (PAS/AB) and viewed in a digital microscope (Olympus, Tokyo, Japan). The pathology in DSS-treated mice was blindly scored for each animal using H&E stained slides, according to the conditions that follow: (i) inflammation severity (0 = none, 1 = mild, 2 = moderate, 3 = severe), (ii) infiltration extent (0 = no infiltrate, 1 = infiltrate around crypt base, 2 = infiltrate reaching to muscularis mucosae, 3 = extensive infiltration reaching the muscularis mucosae and thickening of the mucosa with abundant edema, 4 = infiltration of the submucosa), (iii) epithelial damage (0 = normal morphology, 1 = some loss of goblet cells /some crypt abscesses or damage, 2 = loss of goblet cells in large areas /extensive crypt abscesses or damage, 3 = loss of crypts < 5 crypt widths, 4 = loss of crypts > 5 crypt widths, <20% ulceration, 5 = > 20% ulceration) iv) percentage of epithelial damage, crypt abscessed, crypt loss or ulceration (0 = 0%, 1 = 1–25%, 2 = 26–50%, 3 = 51–75%, 4 = 76–100%). To check mucins and the goblet cells, the sections were stained with Alcian blue and periodic acid–Schiff reagents.

### 4.10. Statistical Analyses

Data are presented as the mean ± SEM. The GraphPad Prism (La Jolla, CA, USA) software program was employed for the analyses and plotting of the data. A nonparametric *t*-test, a one-way ANOVA, followed by Dunnett’s post hoc test, or two-way ANOVA followed by Bonferroni’s post hoc test, were performed wherever applicable to determine statistical differences as indicated in the figure legends.

## Figures and Tables

**Figure 1 ijms-24-00174-f001:**
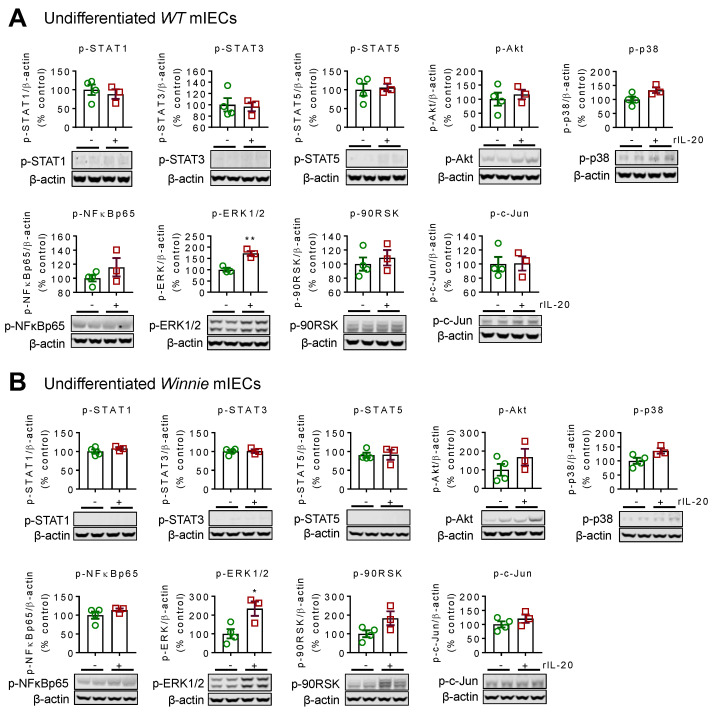
rIL-20-induced signalling pathways in primary intestinal epithelial cells. Representative western blot images of phosphorylated STAT1, STAT3, STAT5, Akt, NF-kβ p65, ERK1/2, 90-RSK, p-c-Jun, and p38 in undifferentiated C57BL/6 (WT) (**A**) and Winnie (**B**) primary colonic organoids following treatment with rIL-20 (100 ng/mL) for 1 h. Data are presented as mean ± SEM with individual cultures from 3–4 independent experiments. * *p* < 0.05 and ** *p* < 0.01 compared with untreated controls (non-parametric Man-Whitney *t*-test).

**Figure 2 ijms-24-00174-f002:**
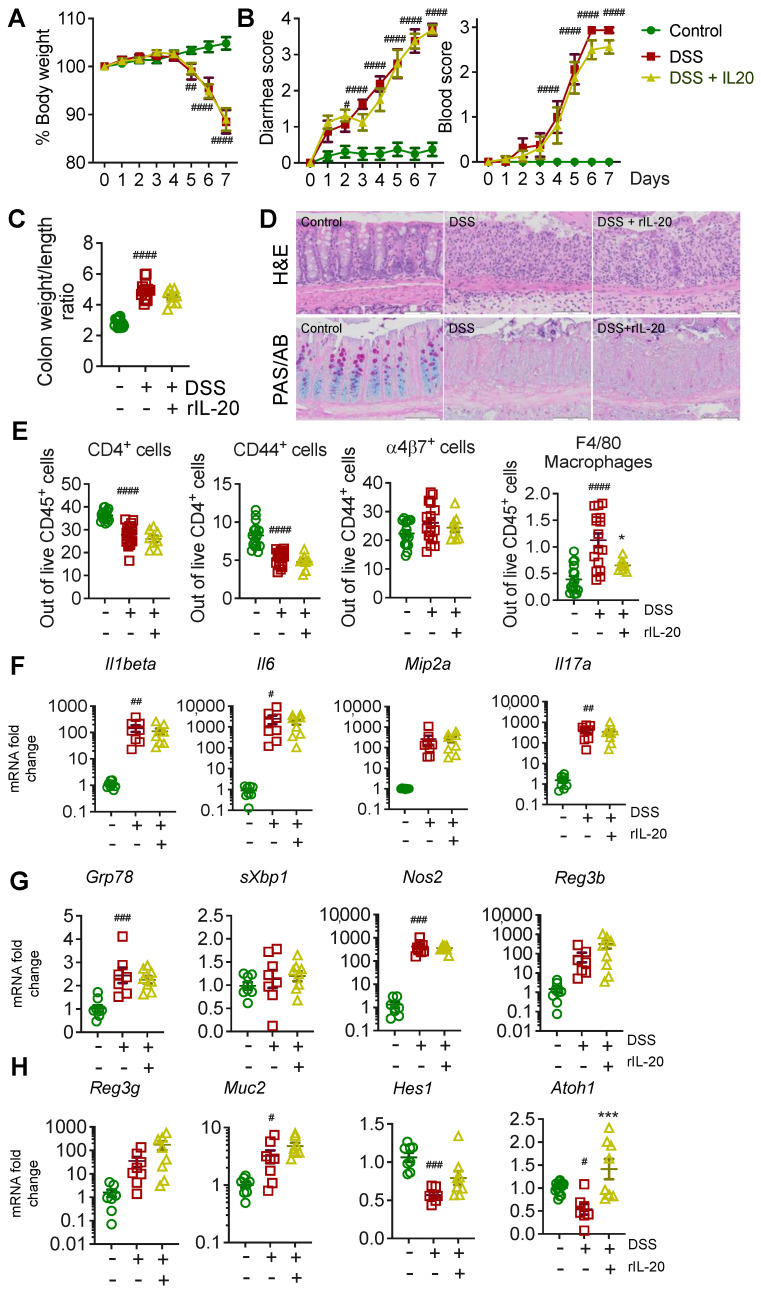
The effects of rIL-20 on DSS-induced colitis. C57BL/6 mice challenged with 2.5% DSS were either treated with rIL-20 (100 ng/g/body weight) or PBS i.p (DSS only). Body weight loss (**A**), diarrhea score (**B**), rectal bleeding score (**C**), colon weight/length ratio (**D**), and representative tissue pathology (haematoxylin and eosin staining) and mucin producing goblet cells in the intestines (PAS/AB staining) (**D**) of the animals. Flow cytometric relative frequency of immune cell populations (**E**), relative gene expression of pro-inflammatory cytokines (**F**) and cellular stress markers (**G**) and antimicrobials (**H**) were measured by qRT-PCR in the primary cells isolated from naïve and rIL-20 treated animals as in B. Data are presented as mean ± SEM (n = 8–12). ^#^
*p* < 0.05, ^##^ *p* < 0.01, ^###^ *p* < 0.001, ^####^ *p* < 0.0001, * *p* < 0.05 and *** *p* < 0.001 when compared with control (One-way ANOVA followed by Bonferroni’s post hoc test).

**Figure 3 ijms-24-00174-f003:**
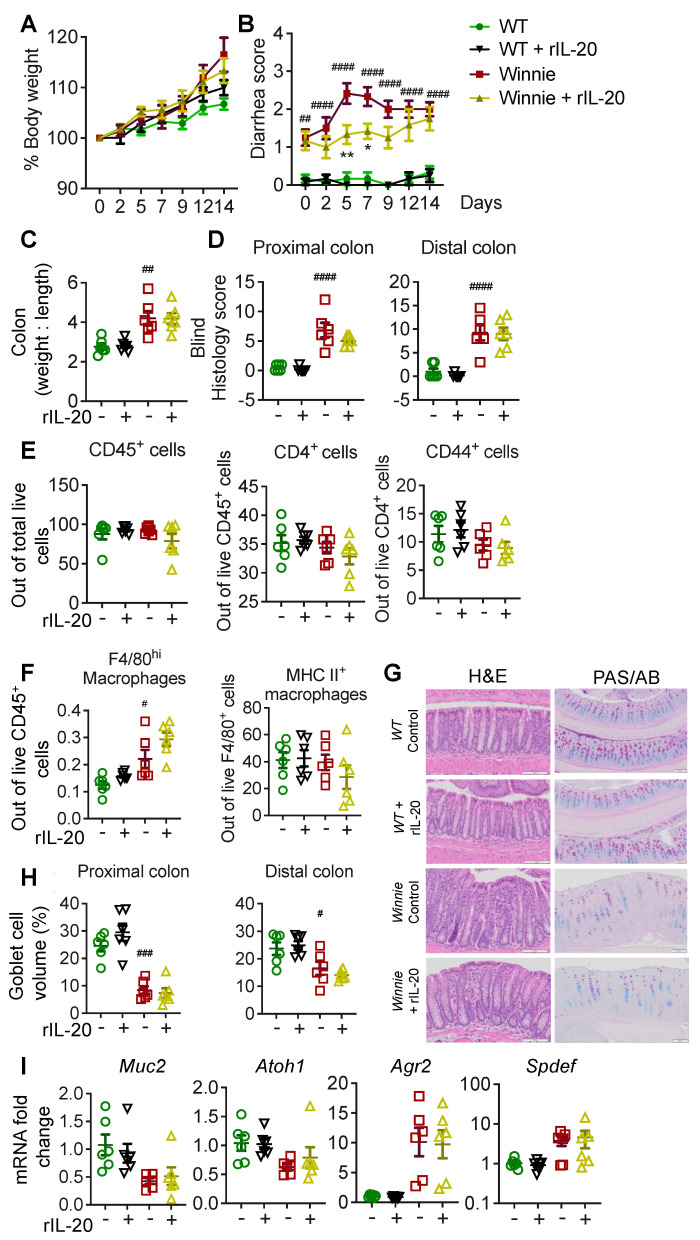
The in vivo effects of rIL-20 on a spontaneous colitis mouse model. C57BL/6 or Winnie mice either treated with rIL-20 (100 ng/g/body weight) or PBS i.p were analysed for body weight loss (**A**), diarrhea score (**B**), colon weight/length ratio (**C**), tissue pathology (**H**,**E**) in the proximal and distal colon (**D**), flow cytometric relative frequency of T cells (**E**), myeloid cells (**F**), representative H&E staining and mucin producing goblet cell (PAS/AB) staining images (**G**), relative colon-specific goblet cell volumes (**H**) and cellular stress markers (**I**) measured by qRT-PCR in the colons of naïve and rIL-20 treated animals as in B. Data are presented as mean ± SEM (n = 6–8). ^#^
*p* < 0.05, ^##^ *p* < 0.01, ^###^ *p* < 0.001, and ^####^ *p* < 0.0001 when compared with control (One way ANOVA followed by Dunnett’s post hoc test).

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
