# Peer review of "IL-20 Activates ERK1/2 and Suppresses Splicing of X-Box Protein-1 in Intestinal Epithelial Cells but Does Not Improve Pathology in Acute or Chronic Models of Colitis"

_ijms, 2022, doi:10.3390/ijms24010174_

Round 1
Reviewer 1 Report
In this study authors have evaluated the role of IL-20 on intestinal epithelial cells and associated signaling cascades and transcriptomic modifications in the murine organoid and murine models. Below are the comments:
1. Whether the authors have looked into other immune cell population (eg.B cells, NK cells etc.) after rIL-20 administration?
2. As mentioned authors are interested to explore the associated signaling cascades and transcriptomic modifications, therefore I would recommend authors to perform Bulk RNA seq analysis either in organoid or in vivo model in presence and absence of rIL-20.
3. What is activation status of ERK1/2 in in vivo models after administration of rIL-20?
4. The authors need to provide a brief summary of their findings, clinical relevance and future implications of this study.
Author Response
Thank you to the reviewer for their comments, we have addressed this below.
In this study authors have evaluated the role of IL-20 on intestinal epithelial cells and associated signaling cascades and transcriptomic modifications in the murine organoid and murine models. Below are the comments:
- Whether the authors have looked into other immune cell population (eg.B cells, NK cells etc.) after rIL-20 administration?
We did not explore the changes in B cells or NK cell population. Our focus was on T cells as these cells mainly drive the pathology in both DSS and Winnie mouse models. However, as shown in our manuscript, no major changes were observed in the animals post rIL-20 treatment as highlighted in the manuscript.
- As mentioned authors are interested to explore the associated signaling cascades and transcriptomic modifications, therefore I would recommend authors to perform Bulk RNA seq analysis either in organoid or in vivo model in presence and absence of rIL-20.
The in-vivo models provide little support for rIL-20-driven changes in the intestine. Furthermore, in-vivo dissecting a direct role of a cytokine treatment is challenging, as it might result in off-target effects. Therefore, to explore the signaling cascade downstream we utilised primary intestinal organoids, which we think provide clearer data. We believe that although interesting, the bulk RNA-Seq approach from in-vivo tissue is outside the scope of the current manuscript.
- What is activation status of ERK1/2 in in vivo models after administration of rIL-20?
This is an interesting question. The ERK1/2 activation is short-lived, assessed after 30 mins of rIL-20 in the ex-vivo organoid system. rIL-20 was given at various times in the in-vivo models, during the experiment and the last administration of rIL-20 prior to the endpoint is more than 24 h beforehand. Therefore, this is hard to assess in-vivo in the models.
- The authors need to provide a brief summary of their findings, clinical relevance and future implications of this study.
We have now elaborated on this and added this to the manuscript. Thank you for the suggestion.
Reviewer 2 Report
This study examines whether IL-20 intraperitoneal injections improves pathogenesis in acute and chronic models of colitis. Ultimately, this study did not show improvement in the mouse models.
I am concerned by the conclusions made that IL-20 treatment could have a possible role in colitis treatment when in fact no evidence of improvement was found in the animals models. While it is appropriate to use in vitro systems to study possible mechanism of the phenotype seen in vivo models, trying to suggest in vitro results upon a negative in vivo study does not make sense.
While IP injections are a more convenient method of parental drug delivery in mouse models, it is not a realistic or reliable method of drug delivery compared to the intravenous route. Further, if such treatment discoveries are to be brought forward to clinical trials, these treatments would be given intravenously. I would highly encourage the authors to repeat their experiments with tail vein injections of IL20. It would be interesting then if positive results are obtained then.
Author Response
Thank you to the reviewer for their comments, we have addressed this below.
This study examines whether IL-20 intraperitoneal injections improve pathogenesis in acute and chronic models of colitis. Ultimately, this study did not show improvements in the mouse models.
I am concerned by the conclusions made that IL-20 treatment could have a possible role in colitis treatment when in fact no evidence of improvement was found in the animals models. While it is appropriate to use in vitro systems to study possible mechanism of the phenotype seen in vivo models, trying to suggest in vitro results upon a negative in vivo study does not make sense.
We appreciate this comment by the reviewer. It was a surprise to us as well, that rIL-20 did not induce protection in the colitis models. However, other cytokines from the same subfamily, like rIL-22 were able to reduce pathology significantly, shown below. Therefore, albeit this is a negative result in-vivo, it is an important finding.
While IP injections are a more convenient method of parental drug delivery in mouse models, it is not a realistic or reliable method of drug delivery compared to the intravenous route. Further, if such treatment discoveries are to be brought forward to clinical trials, these treatments would be given intravenously. I would highly encourage the authors to repeat their experiments with tail vein injections of IL20. It would be interesting then if positive results are obtained then.
We agree that IP injections are convenient in the animal models. However, IV administration of recombinant cytokines is not recommended due to off-target side effects. We have conducted experiments using the subcutaneous route, as this is a more translatable, however observed no alterations in outcome.

Reviewer 3 Report
In the manuscript, the authors have taken steps to tease out and understand the signaling pathway by IL-20 in intestinal epithelial cells and primary intestinal organoid culture with a DSS-induced or mouse model colitis. The authors observed that IL-20 mediated increased phosphorylation of ERK1/2 and reduced tunicamycin-induced ER stress and UPR in the intestinal cells through a significant reduction of spliced XBP1 transcripts upon IL-20 treatment. However, these effects were lost upon differentiation of the intestinal epithelial cells into goblet cells. Further, IL-20 fails to improve the pathology of colitis. The results mentioned above were corroborated in a spontaneous mouse colitis model.
Fig 1: Please include the blots for endogenous Total protein levels, such as Total STAT1/3/5, Total ERK, etc.
Are lanes 1 and 2 and 3 and 4 duplicates of each other? Please label or incorporate the detail in the figure legend.
There is increased phosphorylation of AKT in the fourth lane and of 90R-SK in the third and fourth lanes for undifferentiated Winnie mIECs. How would the authors comment on that?
Fig 2 E and 3F: the F4/80 macrophage infiltration is restricted upon IL-20 treatment in DSS-induced colitis however the opposite is observed in the case of the in-vivo study despite the drop in MHCII-specific macrophages.
The authors have done a beautiful job in studying the improvement of pathology in a colitis model through Il20 treatment. However, their observations fail to reach such conclusions.
Have the authors performed a dose and time-period response with rIL-20. treatment?
How does phosphorylation of ERK relate to the ER stress and UPR response, if any?
Why are the IL-20-mediated protective effects lost upon differentiation of IECs into goblet cells?
The authors mention in the abstract "enhanced expression of ERK1/2"- however, this is not true. The results show an increase in phosphorylation of ERK1/2. Total ERK levels need to be studied to conclude if there is an increase in expression. Which hereon begs the question, why does IL20 phosphorylate ERK only?
Fig 3B- IL-20 treatment provides a transient relief as observed with the faltering diarrhea score until the score post 9 days which presents an interesting result. How does IL-20-mediated protection is brought about? What allows for the transient protection and what genes are upregulated to suppress the early relief seen between 2-9 days? Will a second treatment with another dose of IL-20 allow for a similar relief which could in turn suggest the need for IL-20 treatment in intervals?
Author Response
Thank you to the reviewer for the comments. We have added the comments below.
Fig 1: Please include the blots for endogenous Total protein levels, such as Total STAT1/3/5, Total ERK, etc.
The total protein levels are shown with Beta-actin levels here.
Are lanes 1 and 2 and 3 and 4 duplicates of each other? Please label or incorporate the detail in the figure legend.
Information is now added, each blot contains an n = 2, these are highlighted with the bold black line per blot and stated in the figure legend.
There is increased phosphorylation of AKT in the fourth lane and of 90R-SK in the third and fourth lanes for undifferentiated Winnie mIECs. How would the authors comment on that?
The slight increase in AKT and 90R-SK is observed, however, as we have stated this was not significantly different compared to controls.
Fig 2 E and 3F: the F4/80 macrophage infiltration is restricted upon IL-20 treatment in DSS-induced colitis however the opposite is observed in the case of the in-vivo study despite the drop in MHCII-specific macrophages.
The total F4/80 macrophage infiltration represents the number of macrophages present, however the MHC II+ve macrophages are an indication of the activated macrophages present.
The authors have done a beautiful job in studying the improvement of pathology in a colitis model through Il20 treatment. However, their observations fail to reach such conclusions. Have the authors performed a dose and time-period response with rIL-20. treatment?
This dose of rIL-20 was used as per a comparison to other cytokines from the same family, including IL-22 that are efficacious at this dosage. This is shown as an example here for the reviewer, where IL-22 improved pathology in the DSS model (at the equivalent dose of rIL-20).
How does phosphorylation of ERK relate to the ER stress and UPR response, if any?
There is a correlation of pERK activation in the epithelial cells and the decrease in ER stress. However, this is a little difficult to deduct in the in-vitro model with the use of tunicamycin, as tunicamycin results in several signaling changes in the epithelial cells.
Why are the IL-20-mediated protective effects lost upon differentiation of IECs into goblet cells?
This is due to a downregulation of the IL-20 receptor in goblet cells, whilst IL-20 receptor is highly expressed in the epithelial cells.
The authors mention in the abstract "enhanced expression of ERK1/2"- however, this is not true. The results show an increase in phosphorylation of ERK1/2. Total ERK levels need to be studied to conclude if there is an increase in expression. Which hereon begs the question, why does IL20 phosphorylate ERK only?
Thank you for this comment. We have now changed the wording to state there is increase in pERK1/2, total protein is shown with beta-actin in our studies.
Fig 3B- IL-20 treatment provides a transient relief as observed with the faltering diarrhea score until the score post 9 days which presents an interesting result. How does IL-20-mediated protection is brought about? What allows for the transient protection and what genes are upregulated to suppress the early relief seen between 2-9 days? Will a second treatment with another dose of IL-20 allow for a similar relief which could in turn suggest the need for IL-20 treatment in intervals.
We believe that compared to other cytokines, like rIL-22 which is in the same subfamily as IL-20 (graph attached), rIL-20 is clearly not as effective. Therefore, even though it may interesting to explore the transient improvement, it is unlikely to be translational. Importantly, rIL-20 was administered biweekly in the animals. We speculate that the transient improvement may be due to an inhibition of ER stress in the epithelial cells, but its exploration is out of the scope of this study.

Round 2
Reviewer 1 Report
I would recommend acceptance of this manuscript.
Author Response
Thank you.
Reviewer 2 Report
Thank you for the revisons
Author Response
Thank you.
Reviewer 3 Report
I have no further comments on the paper since the authors have replied to my previous concerns and I am satisfied with the given reply